# JPEG-LM: LLMs as Image Generators with Canonical Codec Representations

## Abstract

Recent work in image and video generation has been adopting the autoregressive LLM architecture due to its generality and potentially easy integration into multi-modal systems. The crux of applying autoregressive training in language generation to visual generation is discretization—representing continuous data like images and videos as discrete tokens. Common methods of discretizing images and videos include modeling raw pixel values, which are prohibitively lengthy, or vector quantization, which requires convoluted pre-hoc training. In this work, we propose to directly model images and videos as compressed files saved on computers via canonical codecs (e.g., JPEG, AVC/H.264). Using the default Llama architecture without any vision-specific modifications, we pretrain JPEG-LM from scratch to generate images (and AVC-LM to generate videos as a proof of concept), by directly outputting compressed file bytes in JPEG and AVC formats. Evaluation of image generation shows that this simple and straightforward approach is more effective than pixel-based modeling and sophisticated vector quantization baselines (on which our method yields a 31% reduction in FID). Our analysis shows that JPEG-LM has an especial advantage over vector quantization models in generating long-tail visual elements. Overall, we show that using canonical codec representations can help lower the barriers between language generation and visual generation, facilitating future research on multi-modal language/image/video LLMs.[1]

## 1 Introduction

With large language models (LLMs) the field of NLP has shifted to multi-task processing (e.g., machine translation, code generation, action planning) using a single LLM with little data needed for adaptation (Ouyang et al., 2022). We envision that future research will continue shifting to multi-modal multi-task processing, where text and visual data are mixed. However, current paradigms of generating images and videos differ substantially from text generation, requiring specialized and complicated training and representations (Van Den Oord et al., 2017; Rombach et al., 2022; Peebles & Xie, 2023). In this work, we simplify the task of image and video generation by using the exact autoregressive transformer architecture as in mainstream LLMs (Radford et al., 2019), over canonical and universal codecs: JPEG for images (Wallace, 1991), and AVC/H.264 for videos (Wiegand et al., 2003).

The key obstacle to training autoregressive models for image and video generation is *discretization*, as continuous data like images and videos need to be represented as discrete tokens. Current generative vision models that follow autoregressive language modeling objectives (Bengio et al., 2000) often adopt vector quantization (VQ) to encode images or videos to some learned latent codes and then apply language models (Van Den Oord et al., 2017; Ramesh et al., 2021; Yu et al., 2021; Yan et al., 2021; Yu et al., 2023).[2] However, VQ methods often demand sophisticated tokenizer training that requires a careful hyperparameter selection for vision-specific modules (e.g., downsampling factor

---

[1]Our code and models will be available at `anonymized`.

[2]The other major line of generative vision models are diffusion models, a score-based, non-autoregressive method (Song & Ermon, 2019; Ho et al., 2020; Rombach et al., 2022; Peebles & Xie, 2023). Since the diffusion objectives are drastically different from the language modeling objective, it is challenging to integrate them in a multi-modal setup (e.g., with regular language models). While not a main focus of this work, we include comparisons with diffusion models in our later experiments as a secondary evaluation.

in convolutions) and balancing across several losses (Van Den Oord et al., 2017; Esser et al., 2021). VQ also involves a two-stage, non-end-to-end learning process (first the neural tokenizer, then the latent code LM). This makes downstream adaptation of the models less flexible (e.g., tuning the VQ tokenizer interferes with the learned latent code LM). Overall, the use of conventional LLM architectures (end-to-end autoregressive sequence modeling) as generative vision models is not yet straightforward.

The seminal work of ImageGPT (Chen et al., 2020) attempted to bridge this gap by using a regular GPT architecture to model pixels sequentially. They have shown a small-scale success at a very low resolution of 32x32 pixels. More realistic images at a size of 256x256 would require modeling a prohibitive amount of tokens in each sequence (65K or 196K tokens depending on color modes), not to mention videos. This hinders the method's wider adoption by the field.

In this work, we tackle the problem of training LLM architectures for image and video generation where the essential discretization neither adds significant complications to the pipeline like VQ methods, nor is computationally prohibitively expensive like ImageGPT. Specifically, we use canonical file encodings/codecs—JPEG for images (Wallace, 1991), and AVC/H.264 for videos (Wiegand et al., 2003)—as non-neural preprocessors that discretize data. We show that codec-based representations greatly mitigate the sequence length limitation while being simple and effective. This design enables us to train a vanilla transformer with the conventional language modeling objective for image and video generation in a realistic setup.

We pretrain two 7B models with a Llama-2 architecture (Touvron et al., 2023), named JPEG-LM and AVC-LM, that can generate 256x256 images and 256x144 videos with 15 frames, with an average context length of 5K and 15K, respectively. In our main image modeling/generation evaluations, we show that JPEG-LM surpasses strong VQ-based models in generation quality (an average of 31% FID reduction) and produces surprisingly realistic qualitative examples. Our results also show AVC-LM can generate videos with realistic movements. Furthermore, we analyze in which aspects JPEG-LM is particularly stronger than VQ models and discover that our non-neural, training-free codec representations are more competent in capturing long-tail elements in images (e.g., human faces/eyes and text characters in small sizes).

Overall, this work presents how conventional LLM architectures can be used as generalized models towards visual generation. Our approach using canonical codecs does not incur vision-specific complications in the pipeline or suffer from sequence length infeasibility seen in prior work. Compared to the baselines, our models are much simpler to train and more effective. Following the previous efforts in unifying detached language-based tasks, our method helps pave the way to a unification of multiple modalities, facilitating the exploration of porting LLM techniques (e.g., alignment, scaling, efficiency, security, etc.) to all modalities.

## 2 BACKGROUND

In this work, we explore autoregressive image generation as a straightforward extension of prominent LLM setups (Radford et al., 2019).[3] Conventional language modeling (Bengio et al., 2000) models the likelihood of sequential data autoregressively. Specifically, given a sequence of discrete tokens $x_1, x_2, \cdots, x_N$ (or $\boldsymbol{x}_{1:N}$), a language model models $p(\boldsymbol{x}_{1:N}) = \prod_{i=1}^{N} p(x_i \mid \boldsymbol{x}_{1:i-1})$, an objective used in most mainstream LLMs. The key of applying language modeling to visual generation is how to discretize continuous data $\boldsymbol{x}$ like images and videos to discrete tokens $\boldsymbol{x}_{1:N}$ like in language. Below we give an overview of two prominent approaches to the discretization of images.

### 2.1 PIXEL VALUES: IMAGEGPT

ImageGPT (Chen et al., 2020) is an image generation model based on a conventional LLM architecture (GPT-2). The images are discretized as a sequence of pixel values (integers 0–255) from the upper-left to the bottom-right pixel (raster scan). Since there are three channels of colors for each pixel, to reduce the number of tokens in each pixel sequence, ImageGPT clusters pixel colors to 512 distinctive clusters (i.e., for each pixel, three values from 0 to 255 are converted to one value from 0 to 511).

---

[3]As a proof of concept, we mainly explore autoregressive modeling in visual generation only (images and videos, without text-conditioning), while future work may explore more diverse multi-modal setups.

ImageGPT models the probability of pixel sequences autoregressively: $p(\text{pixel-value}(\boldsymbol{x})_i \mid \text{pixel-value}(\boldsymbol{x})_{1:i-1})$. This is an expensive process, and ImageGPT only models and generates 32x32 images. Images with a more realistic resolution like 256x256 would require 65K tokens for each image (or 196K tokens without color clustering), a prohibitive sequence length for LLMs.

## 2.2 LATENT CODES: VECTOR-QUANTIZATION MODELS

Vector-quantization (VQ) operates as a two-stage process, tokenizer training and language model training (Esser et al., 2021; Ramesh et al., 2021). We take VQ-VAE as our example tokenizer which discretizes continuous images (Van Den Oord et al., 2017). The tokenizer first learns an encoder $E$ to project an image $\boldsymbol{x}$ to spatial features $E(\boldsymbol{x})$. Then for each feature $\boldsymbol{e}$ in $E(\boldsymbol{x})$, it is quantized to $\hat{\boldsymbol{z}}$ by looking up the nearest neighbor in a learned codebook $\mathcal{Z}$: $\hat{\boldsymbol{z}} = \text{quantize}(E(\boldsymbol{x})) = [\arg\min_{\boldsymbol{z}_k \in \mathcal{Z}} \|\boldsymbol{e} - \boldsymbol{z}_k\|_2^2]_{\boldsymbol{e} \in E(\boldsymbol{x})}$. The index $k$ of the nearest entry in codebook $\mathcal{Z}$ for each spatial feature forms the sequence of VQ latent codes. A decoder $G$ is then learned to reconstruct the original image from the quantized representations. Overall, VQ-VAE learns an encoder $E$, decoder $G$, and codebook $\mathcal{Z}$, with three distinct losses: reconstruction loss, codebook loss, and commitment loss. $L_{\text{VQ-VAE}} = \|\boldsymbol{x} - G(\hat{\boldsymbol{z}})\|_1 + \|\text{sg}[E(\boldsymbol{x})] - \hat{\boldsymbol{z}}\|_2^2 + \beta\|\text{sg}[\hat{\boldsymbol{z}}] - E(\boldsymbol{x})\|_2^2$. An effective VQ tokenizer needs a large amount of training data, proper hyperparameters for the vision-specific modules (e.g., downsampling factor in convolutional encoder $E(\cdot)$), and a careful balance between the different losses (e.g., in $L_{\text{VQ-VAE}}$), which add significant complications to the pipeline.

A language model architecture can then be trained over the VQ latent codes (a sequence of index $k$ above) as a generative vision model: $p(\text{VQ-code}(\boldsymbol{x})_i \mid \text{VQ-code}(\boldsymbol{x})_{1:i-1})$. Notably, since the training of language model comes after and depends on the VQ tokenizer, a post-hoc update to the VQ tokenizer is challenging since it would lead to a non-trivial retraining or adaptation of the trained language model. Indeed in §5.3 we find that the VQ tokenizer, though trained with a large amount of data, still struggles with long-tail elements in the images and is hard to be optimized once and for all.

For simplicity and end-to-end adaptability, we propose to discretize continuous image and video data via canonical codecs.

## 3 JPEG-LM AND AVC-LM

Though images and videos are continuous data and naturally have 2D or 3D data structures, they are stored as files on computers efficiently via compression/codecs, which leads to a discrete 1D representation. We aim to explore whether standard LLM architectures can directly learn to model and generate canonical vision file encodings, which can subsequently be read/opened as generated images or videos. Generation in this paradigm would greatly mitigate the sequence length infeasibility in ImageGPT while being simple and end-to-end trainable compared to VQ methods. Moreover, canonical file encodings/codecs are often non-neural and training-free and are robust to distributional shifts (§5.3). In this work, we choose the most popular and established file encodings/codecs for images and videos, JPEG (Wallace, 1991) and AVC/H.264 (Wiegand et al., 2003), respectively.[4]

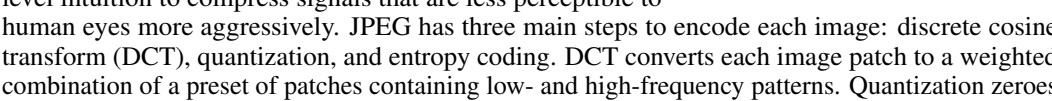

Figure 1: JPEG-LM and AVC-LM are simple autoregressive transformers that directly model and generate canonical file encodings.

### 3.1 CANONICAL CODECS: JPEG AND AVC/H.264

Canonical non-neural codecs like JPEG and AVC have a high-level intuition to compress signals that are less perceptible to human eyes more aggressively. JPEG has three main steps to encode each image: discrete cosine transform (DCT), quantization, and entropy coding. DCT converts each image patch to a weighted combination of a preset of patches containing low- and high-frequency patterns. Quantization zeroes

---

[4]For images, PNG is also a common format. However, unlike the lossy JPEG, PNG is a lossless compression method (similar to ZIP) and often results in less effective compression and much longer sequences than JPEG.

out some high-frequency patterns from the weighted combination, since human eye is not good at perceiving them. Entropy encoding such as Huffman coding is then used to reduce the total numbers/bits representing the patches/images.[5]

AVC (H.264) operates on patches (macroblocks) of video frames. Each patch can be encoded using blocks of pixels that are already encoded within the current frame (intra-frame prediction) or using blocks of pixels encoded in other frames (inter-frame prediction with motion estimation). The prediction is then subtracted from the current patch to form a residual. The residual then goes through a process similar to JPEG, involving DCT, quantization, and bitstream encoding. The encoded content is a crucial part to the subsequent container files like MP4.

Both codecs have been used widely for decades and substantially compress the data (and thus sequence length) compared to raw pixel modeling (in our setup 40x in JPEG and 110x in AVC). Our focus is to use these canonical codecs as off-the-shelf tools to convert images and videos to sequences of discrete bytes efficiently.[6] We wish to fit an LLM to implicitly learn the grammars and semantics of the canonical codecs.

## 3.2 JPEG-LM AND AVC-LM

JPEG and AVC convert images and videos to bytes. Most of these bytes represent the image and video content after entropy encoding. However, there are also metadata and special patch/macroblock separators that are invariant across images or videos and use up multiple bytes. To address them along with other unknown frequent byte combinations that are compressed suboptimally by entropy encoding (e.g., by JPEG's standard, fixed Huffman tables), we further extend the default byte vocabulary (256 discrete values) *slightly* with byte-pair encoding (BPE), a standard preprocessing scheme in LLMs, which merges bytes appearing together frequently to a new single token.[7] Since JPEG and AVC produce sequences of variable lengths based on the content of images and videos, special beginning-of-sequence and end-of-sequence tokens are also added to the vocabulary. The entries in the vocabularies are considered as our JPEG/AVC tokens.

Given an image $\boldsymbol{x}$, we propose JPEG-LM to model $p(\text{JPEG-token}(\boldsymbol{x})_i \mid \text{JPEG-token}(\boldsymbol{x})_{1:i-1})$. Given a video $\boldsymbol{x}$, we propose AVC-LM to model $p(\text{AVC-token}(\boldsymbol{x})_i \mid \text{AVC-token}(\boldsymbol{x})_{1:i-1})$. We use conventional LLM architectures (autoregressive transformers) without any vision-specific modifications (no convolutions, no 2D positional embeddings) to maximize the models' generality.

## 4 EXPERIMENTAL SETUP

### 4.1 JPEG-LM

We pretrain a 7B Llama-2 model (Touvron et al., 2023) from scratch using 23M 256x256 images subsampled from Schuhmann et al. (2022). JPEG encodes each image with a quality factor of 25 (qualitative illustration in §5.3).[8] We first use 10K images to derive 320 BPE tokens as our vocabulary entries.[9] On average, each image in our training data leads to 5K tokens. For batching efficiency, we concatenate all sequences in the dataset and chunk in sequences of length 12K. In total, we have 9.5M sequences and thus 114B JPEG tokens (for each epoch). The model is trained approximately for two epochs with a maximum learning rate of 3e-4.

---

[5]A further intuitive and interactive description can be found at `https://parametric.press/issue-01/unraveling-the-jpeg/` (Shehata & Conlen, 2019).

[6]Both codecs operate at bits level at the core (due to entropy encoding), but modeling at bytes level is effective according to our experiments.

[7]More precisely, for the metadata/headers in the byte sequence that are well-known to be redundant across examples (e.g., JPEG quantization and Huffman tables), we remove them in the preprocessing and later add them back to the generated bytes from the model. For more complicated codecs like AVC, we let BPE handle such metadata.

[8]`https://pillow.readthedocs.io/`

[9]In our pilot study, we find the BPE process to be optional and the model would work similarly without it. The 64 extended vocabulary entries apart from the 256 default byte values include special JPEG separators FFD0, FFD1, . . . , FFD9, FFDA, and static file headers invariant across data, which slightly help reduce the sequence length. The vocabulary size 320 is chosen since a multiple of 64 for the embedding dimension is desired for optimal compute on GPUs.

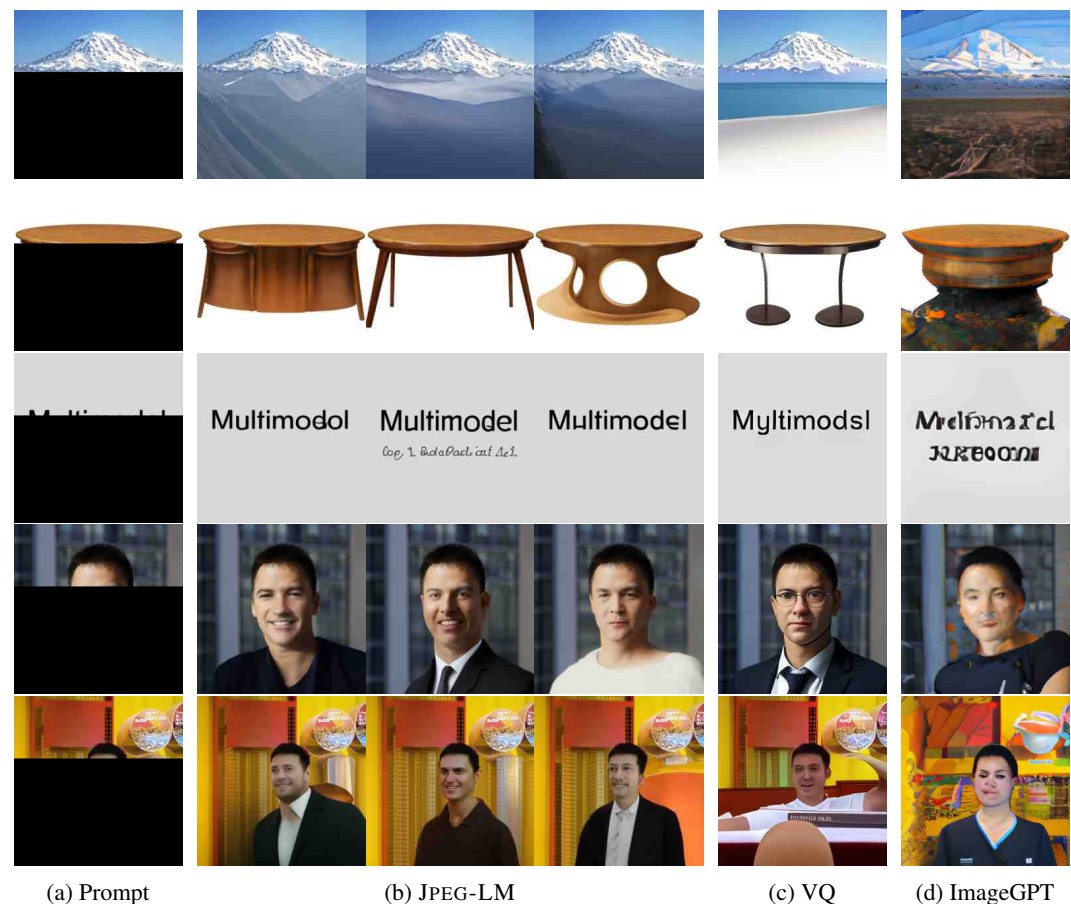

| (a) Prompt | (b) JPEG-LM | (c) VQ | (d) ImageGPT |

Figure 2: Generated images by JPEG-LM and baselines with partial images as prompts. We show three random samples from JPEG-LM and one from VQ transformer and ImageGPT (with super-resolution). The original images for the prompts are independently sourced outside existing training sets. We observe that JPEG-LM can generate realistic facial expressions, landscape, common objects, texts in image forms, etc. Additionally, JPEG-LM shows an especial advantage over baselines on meaningful details like human eyes (zoom in for the best view). Figure 6 and Figure 7 show further examples of JPEG-LM and VQ transformer on unconditional generation.

### 4.2 AVC-LM

As a proof of concept that canonical video codecs can be used for video generation as well, similar to JPEG-LM, a 7B Llama-2 model is pretrained from scratch as AVC-LM using 2M 256x144 videos subsampled from Bain et al. (2021). Due to the scope of experiments, we only keep the first 5 seconds of each video with 3 frame-per-second (thus 15 frames in total). The video is then processed with AVC/H.264 codec with a constant quantization parameter 37.[10] We use 10K videos to derive 1024 BPE tokens as the vocabulary entries. On average, each video in our training data has 15K tokens. We perform data concatenation and chunk in context lengths of 32K for efficient batching. In total, we have 1.3M sequences and thus 42B AVC tokens.

### 4.3 IMAGE GENERATION BASELINES

**VQ transformer** We use a pretrained VQ tokenizer from Tang et al. (2022), which used 200M images (ITHQ-200M, closed source dataset) to train a VQ-VAE model.[11] This VQ tokenizer processes each image in the 23M image training set for our JPEG-LM (vocabulary size 4096, sequence length

---

[10]https://ffmpeg.org/

[11]In our pilot study, we found this f8 VQ tokenizer outperforming other open-source VQ tokenizers, even the ones with longer context lengths (f4) like in Rombach et al. (2022). More discussion can be found in §A.

1024). We then train a 7B Llama-2 transformer with the same configuration as in JPEG-LM. We use this VQ model as a main comparison to our JPEG-LM throughout this work.

**ImageGPT + super-resolution** ImageGPT uses GPT-2 XL as its underlying architecture. The pretrained model in (Chen et al., 2020) is trained over 14M 32x32 images from ImageNet. For a comparable evaluation, we use a super-resolution model (Rombach et al., 2022) over ImageGPT's output.[12]

**Diffusion** Though not a focus of this work, we include two variants of diffusion models in the baselines, Stable Diffusion (inpainting optimized) (Rombach et al., 2022) and VQ diffusion (Gu et al., 2022; Tang et al., 2022). Both diffusion models can take partial images (through masking) and generate completed images, a setup we use across models in later evaluations. These baseline diffusion models are smaller in model size (~1B) but consume orders of magnitude more training data (200M–5B). They only serve as a secondary reference, and our focus is on comparing autoregressive image generation models under mainstream LLM paradigms.

## 5 RESULTS

In works of language modeling, a fundamental evaluation is to collect a set of validation data, use the prefixes of data as prompts to the pretrained language model, and sample from the language model for a completion (Holtzman et al., 2020; Meister et al., 2023). The completions are then evaluated for their quality against the gold validation data through distance metrics like Mauve score (Pillutla et al., 2021).

In this work, since we focus on vision-modality-only models with LLM architectures, we retain partial images (and later partial videos) as prompts to our models and evaluate their completions. Given a prompt ratio $r_{\text{prompt}}$, the autoregressive image generation models condition on $\text{discretization}(\boldsymbol{x})_{1:(r_{\text{prompt}} \times N_{\text{tokens}})}$ for the generation.[13] Throughout the eval-

Table 1: Zero-shot, partial-image-conditioned, FID evaluation on **ImageNet-1K** (lower is better). $r_{\text{prompt}}$ indicates the ratio of the image passed to the model as prompt. Best results among the autoregressive models are in bold fonts (reference diffusion results are italicized if better).

| | $r_{\text{prompt}} = 0.25$ | $r_{\text{prompt}} = 0.5$ | $r_{\text{prompt}} = 0.75$ |
|---|---|---|---|
| Stable Diffusion (inpaint) | 266.71 | 132.98 | 58.17 |
| | ($\pm$1.67) | ($\pm$0.53) | ($\pm$0.10) |
| VQ Diffusion | *252.42* | 125.16 | 57.49 |
| | ($\pm$0.20) | ($\pm$0.26) | ($\pm$0.25) |
| ImageGPT (super-resolution) | 289.48 | 262.76 | 258.11 |
| | ($\pm$0.61) | ($\pm$0.48) | ($\pm$0.69) |
| VQ Transformer | 302.92 | 172.73 | 71.88 |
| | ($\pm$0.29) | ($\pm$0.21) | ($\pm$0.19) |
| JPEG-LM | **272.12** | **123.09** | **34.21** |
| | ($\pm$1.24) | ($\pm$0.28) | ($\pm$0.21) |

Table 2: Zero-shot, partial-image-conditioned, FID evaluation on **FFHQ** (lower is better). $r_{\text{prompt}}$ indicates the ratio of the image passed to the model as prompt. Best results are in bold fonts. The prompting ratios in FFHQ were chosen differently such that they often lead to image prompts above the human eyes, below the eyes, and below the nose in pilot experiments.

| | $r_{\text{prompt}} = 0.375$ | $r_{\text{prompt}} = 0.4375$ | $r_{\text{prompt}} = 0.5$ |
|---|---|---|---|
| Stable Diffusion (inpaint) | 115.30 | 107.02 | 89.82 |
| | ($\pm$2.14) | ($\pm$1.83) | ($\pm$4.51) |
| VQ Diffusion | 60.88 | 45.63 | 40.58 |
| | ($\pm$0.38) | ($\pm$0.17) | ($\pm$0.91) |
| ImageGPT (super-resolution) | 61.73 | 57.80 | 55.28 |
| | ($\pm$0.91) | ($\pm$0.73) | ($\pm$1.22) |
| VQ Transformer | 53.25 | 45.58 | 41.15 |
| | ($\pm$0.54) | ($\pm$0.58) | ($\pm$0.35) |
| JPEG-LM | **36.15** | **31.22** | **27.15** |
| | ($\pm$1.11) | ($\pm$0.33) | ($\pm$0.21) |

Table 3: Unconditional FID comparison of JPEG-LM and VQ transformer.

| VQ Transformer | 155.51 | JPEG-LM | **121.35** |
|---|---|---|---|
| | ($\pm$2.41) | | ($\pm$0.51) |

---

[12]The pretrained model provides 4x super-resolution. In our pilot study, we find performing a 4x super-resolution, followed by a 0.5x downsample, then another 4x super-resolution yields the best result for the $32^2$-to-$256^2$ conversion.

[13]More specifically, the fixed-length VQ transformer and ImageGPT condition on $\text{discretization}(\boldsymbol{x})_{1:(r_{\text{prompt}} \times N_{\text{tokens}})}$ and generate $\text{discretization}(\boldsymbol{x})_{(r_{\text{prompt}} \times N_{\text{tokens}}):N_{\text{tokens}}}$. Variable-length

uations, the comparison between JPEG-LM and VQ transformer would be the most direct, as they share the same paradigm, model size, and training data (except that VQ transformer uses substantially more data in the tokenizer training stage).

## 5.1 QUALITATIVE ANALYSIS

In Figure 2, we show the generation samples from JPEG-LM along with baseline models over independently sourced data outside existing training sets. We observe that by directly outputting JPEG file bytes, JPEG-LM can generate surprisingly realistic facial expressions (especially the eyes, compared to the strong VQ transformer), landscape, common objects, texts in image forms, etc. Figure 6 and Figure 7 show examples of JPEG-LM and VQ transformer on unconditional generation.

## 5.2 QUANTITATIVE RESULTS

In Table 1, we show prompting JPEG-LM, VQ transformer, and other baselines with different levels of partial images in ImageNet-1K (Russakovsky et al., 2015). The FID evaluation (Heusel et al., 2017) contains 5000 randomly sampled images from ImageNet-1K's validation set. This is *zero-shot* generation (w.r.t. models' training sets) and without class-conditioning. Experiments were done three times with different seeds. JPEG-LM consistently outperforms the VQ transformer in all prompting ratios. It mostly surpasses diffusion baselines with inpainting capabilities as well.

In Table 2, we show prompting the models with partial images in FFHQ (Karras et al., 2019). This is also a *zero-shot* setup without training to the FFHQ distribution and is evaluated on 1000 randomly sampled FFHQ images. JPEG-LM consistently outperforms the VQ transformer and other baselines.

In Table 3, we further validate our findings on fully unconditional generation with JPEG-LM and VQ transformer. Since they were trained on the same training data, we can compare their FID of unconditional generation w.r.t. our held-out, i.i.d. evaluation set. We again observe that JPEG-LM achieves a better FID.

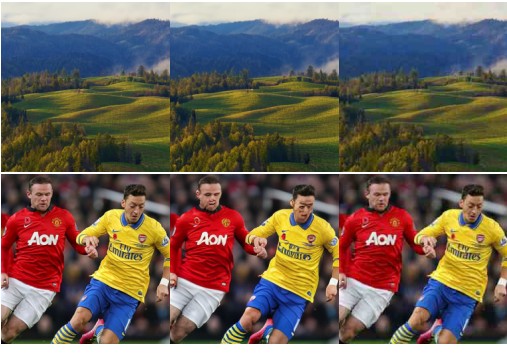

(a) Original     (b) After VQ     (c) After JPEG

Figure 3: Compression effect of VQ and JPEG (zoom in for the best view). JPEG is significantly better in detailed but highly perceptible elements like small human faces and text characters. VQ has a relative advantage in color and sharpness preservation.

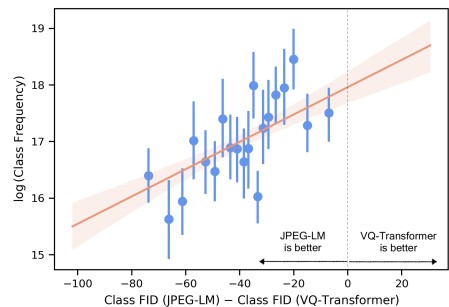

Figure 4: Correlation between per-class (ImageNet-1K) FID difference and class frequency. The class frequency is estimated through querying Google image search. Each class has a corresponding data point while an aggregation is performed for visual clarity. The correlation is positive and statistically significant ($p$=0.0002). This indicates JPEG-LM has more advantage in long-tail classes.[14]

These findings show JPEG-LM's overall competence in image generation with a pure LLM architecture modeling canonical file encodings.

---

JPEG-LM conditions on discretization$(\boldsymbol{x})_{1:\text{patch-position}(r_{\text{prompt}} \times N_{\text{patches}})}$ and generates until a EOS token is produced. Throughout the work, sampling from autoregressive transformers is by default with top-$p = \{0.9, 1.0\}$ and top-$k = \{40, 80\}$.

[14]To further corroborate our findings, apart from using Google image search, we also estimate the class frequency by matching class names/descriptions to the captions of our training images. The correlation is again positive and statistically significant (slope=+106.7, $p$-value=0.006).

### 5.3 WHY JPEG-LM? A CASE STUDY OVER LONG-TAIL ELEMENTS IN IMAGES

To further explore in which aspects our JPEG-LM excels compared to the baselines, especially the VQ transformer, we first compare how data is processed/compressed before being trained in transformers in JPEG-LM and VQ models.

**JPEG vs. VQ compression** JPEG-LM and VQ transformers can both be interpreted as first performing compression and then autoregressive modeling. The VQ model, unlike the non-neural JPEG compression, trained its VQ-VAE quantizer with a large amount of data (200M images in our case). In Figure 3, we observe that both compression methods are relatively successful in compressing and decompressing general scenes like nature/landscape backgrounds. However, we find VQ suffers in small but highly perceptible elements in the images, like human faces or eyes. For images that contain small text characters, we observe the image degradation in VQ also happens in a non-predictable way, generating seemingly clear but uninterpretable text characters. On the other hand, the image degradation due to the non-neural, training-free JPEG compression happens in a predictable manner, arguably more preferable, especially when images contain long-tail elements with important meanings.

**Quantitative analyses on long-tail elements** In Figure 4, we first show the per-class FID in our ImageNet-1K generation experiments. For each class of images, we calculate the difference between their FID with JPEG-LM generations and FID with the VQ transformer generations. We also estimate the frequency/coverage of each class of images on the internet by querying Google image search and logging the total number of returned results. We observe a statistically significant correlation between the per-class FID difference and the class frequency. The more advantage we observe in JPEG-LM over the VQ model, the less frequent the corresponding class is. In other words, JPEG-LM excels relatively more in long-tail sub-distributions.

Table 4: Zero-shot, partial-image-conditioned, FID evaluation on **downscaled FFHQ** (for both FID and $\Delta$, lower is better). An increased gap between JPEG-LM and the VQ transformer shows JPEG-LM is more robust to small but meaningful long-tail elements.

| | $r_{\text{prompt}} = 0.375$ | $r_{\text{prompt}} = 0.5$ |
|---|---|---|
| Stable Diffusion (IP) | 136.28 (±2.48) | 120.54 (±6.46) |
| $\Delta_{\text{downscaled−original}}$ | +20.98 | +30.72 |
| VQ Diffusion | 83.63 (±1.16) | 47.90 (±1.12) |
| $\Delta_{\text{downscaled−original}}$ | +22.75 | +7.32 |
| ImageGPT (SR) | 46.67 (±0.62) | 40.46 (±0.70) |
| $\Delta_{\text{downscaled−original}}$ | −15.06 | −14.82 |
| VQ Transformer | 56.33 (±0.86) | 47.94 (±0.21) |
| $\Delta_{\text{downscaled−original}}$ | +3.08 | +6.79 |
| JPEG-LM | **35.80** (±0.17) | **26.25** (±0.45) |
| $\Delta_{\text{downscaled−original}}$ | −0.35 | −0.90 |

In Table 4, we further intervene on the FFHQ images by downsizing them (to 0.5x, while padding the images with black background to keep the overall size), aiming to test different models' performance on smaller visual concepts (e.g., small human faces). Such concepts, though small in size, can still be highly perceptible by humans and contain important meanings. We thus want the models to be robust on them. We perform similar prompted image generations with JPEG-LM, VQ transformer, and other baseline models.[15] We find that JPEG-LM still consistently outperforms the VQ transformer (and other baselines as well). Especially, JPEG-LM achieves slightly better performance while VQ transformer becomes worse compared to the experiments with original image size. These deltas in opposite directions highlights the robustness of JPEG-LM.

These findings show that JPEG-LM not only has a promising performance overall, but specially strong with long-tail visual elements in the images.

### 5.4 PROOF-OF-CONCEPT VIDEO GENERATION

One advantage of using canonical file encodings in LLM paradigms for vision generation is simplicity. From JPEG-LM that generates images, we naturally take one step further and train a video generation model, AVC-LM, that models canonical video codecs (AVC/H.264) with autoregressive transformers.

---

[15]The FID is measured on the active proportion of the images, excluding the black paddings.

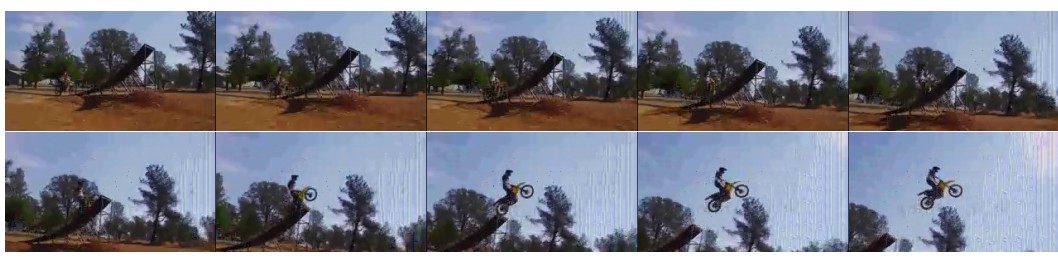

(a) Prompt frames

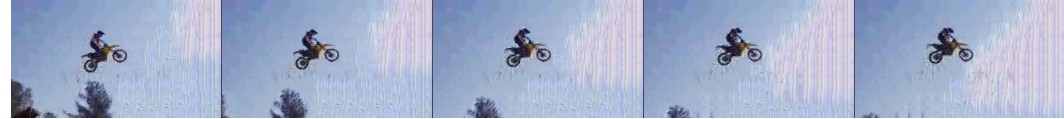

(b) Generated frames

Figure 5: Generated video frames by AVC-LM on held-out test data. The first 10 frames are given to the model as the prompt, and the last 5 frames are generated by the model.

As a proof of concept, we prompt AVC-LM with partial videos (i.e., frames) from a held-out set from our training data and investigate the model completions. In Figure 5 (along with §C), we show qualitative examples generated by AVC-LM. We observe that AVC-LM can capture the motion of moving objects reasonably.

## 6 RELATED WORK

Current image and video generation models often adopt an autoregressive or diffusion approach. The autoregressive approach can build upon pixel-based representations as explored in Van Den Oord et al. (2016); Van den Oord et al. (2016); Chen et al. (2020). These methods suffer from prohibitively long sequences and only operate on low-resolution images. The autoregressive approach can also build upon vector quantization, which involves a sophisticated pre-hoc tokenizer training in addition to the autoregressive model (Van Den Oord et al., 2017; Esser et al., 2021; Ramesh et al., 2021; Yu et al., 2021; Yan et al., 2021; Yu et al., 2023; Mentzer et al., 2023; Lu et al., 2023; Liu et al., 2024a). Diffusion models generate images or videos by an iterative denoising process, and they have specialized objectives and architectures that are challenging to be incorporated to regular LLM paradigms to form multi-modal systems (Song & Ermon, 2019; Ho et al., 2020; Rombach et al., 2022; Ho et al., 2022; Gu et al., 2022; Tang et al., 2022; Gu et al., 2023; Peebles & Xie, 2023; Crowson et al., 2024). For example, performing simple tasks outside visual generation like classification with diffusion architectures is already not straightforward (Li et al., 2023). In this work, we propose to model canonical codecs (JPEG and AVC/H.264) with conventional language model architectures for visual generation. Horton et al. (2023) and Wu et al. (2024) are independent work that also process file bytes data, but they both focus on visual understanding (instead of generation) and use specialized modules to handle the byte sequences (whereas we use a general Llama-2 model). Perez et al. (2024) concurrently discover that JPEG formats can be used with language models in file anomaly handling and generation (on low-resolution images). As a universal codec, JPEG is a novel form of data encoding for efficient image understanding (Park & Johnson, 2023). Kang et al. (2019) explore an image generation model that performs generation and JPEG compression in one system with GANs. JPEG artifacts can also be corrected by learning a restoration model (Kawar et al., 2022), which is potentially helpful to the generations from our JPEG-LM for improving image quality. Compressive codecs are also a rising topic in language. Jiang et al. (2023) use canonical compressors as feature extractors for texts. Lester et al. (2024) train language models to generate compressed texts.

## 7 CONCLUSION

In this work, we propose JPEG-LM and AVC-LM that generate images and videos using mainstream LLM architectures (autoregressive transformers) with canonical codec representations (JPEG for

images, AVC/H.264 for videos). Our approach greatly mitigates the length infeasibility of pixel-based sequence modeling while enabling simple, flexible, and end-to-end training compared to sophisticated vector quantization methods. Our image generation evaluation shows JPEG-LM achieves better results than the baselines, with an especial advantage in generating long-tail visual elements. Our work contributes to a unifying paradigm of language generation and visual generation, facilitating future research to port successful LLM techniques (e.g., alignment, efficiency, etc.) to all modalities.

One notable significance of this work is to show that vanilla autoregressive language modeling with canonical codecs is *indeed possible* in visual generation. This is an approach almost void of prior work, likely because there are many potential, assumed challenges with the method. For example, both JPEG and AVC operate at bits level due to the entropy coding. The bytes in the files do not have consistent meanings and would depend on their context and the implicit Huffman tables. For generality, our models also do not use any vision-specific modules like convolutions or 2D positional embeddings, potentially making the task more challenging. However, we observe that conventional, vanilla language modeling surprisingly conquers these challenges without special designs as training goes. Based on the findings of this work, future work may continue to investigate the scaling aspect of this family of models (similar to mainstream LLMs), co-training/deployment with text-based LLMs, or better architectures for canonical codecs without loss of generality for other modalities. An extended discussion can be found in §A.

## LIMITATIONS

Machine learning models that generate images, especially the models using natural language as convenient controls or even deepfakes that are maliciously trained to swap faces, lead to risks of generating unsafe and harmful content (Nguyen et al., 2022; Qu et al., 2023). Though we mitigate such risks in our model by not including texts for conditioning and not processing multiple images/videos for any types of synthesis, the use cases of the model still require extensive care. The purpose of this work is purely scientific—to explore a fundamental algorithm for general visual generation. Our approach helps lower the barriers of porting LLM techniques to visual generation, and we plan on adopting advances in LLMs (e.g., alignment and watermarking) to further enhance safety in future work (Ganguli et al., 2022; Kirchenbauer et al., 2023). In this work, we pretrain a 7B model. Even with our moderate-scale data, we estimate a full training of JPEG-LM to take a month on 32 Nvidia A100 GPUs. As our model shares the same architecture as regular LLMs, we plan on exploring techniques in LLM efficiency to reduce compute footprint in future work.

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

## A    CONTINUED DISCUSSION

Our work focuses on the challenging task of visual generation (e.g., outputting images) rather than visual understanding (e.g., inputting images, outputting classes or texts). In the field of visual understanding, the encoding of images has less restricted forms. For example, Bavishi et al. (2023) and El-Nouby et al. (2024) linearly project image patches as inputs to the transformers, Liu et al. (2024b) pass CLIP embeddings (Radford et al., 2021) to language models, etc. However, these image encoding formulations are not applicable to image generation. Though not a focus in this work, future work may extend our JPEG-LM and AVC-LM that share the same underlying architectures with regular language models to image and video understanding scenarios.

Compared to raw pixel modeling that would represent a 256x256 image with 65K or 196K tokens (depending on color modes), using canonical codecs like JPEG substantially reduces the sequence length to 5K on average. In terms of sequence length, the VQ transformers are usually more aggressive, representing each image with 1K tokens. It is notable that this an ideal hyperparameter discovered in prior work that helps model global structures—increasing the number of tokens in VQ (thus reducing the downsampling patch size) may lead to degenerated results rather than helping the model learn with more capacity (Esser et al., 2021). Our work proposes to model canonical codecs as a proof of concept, and future work may compare with more VQ setups or further improve the context efficiency of JPEG-LM.

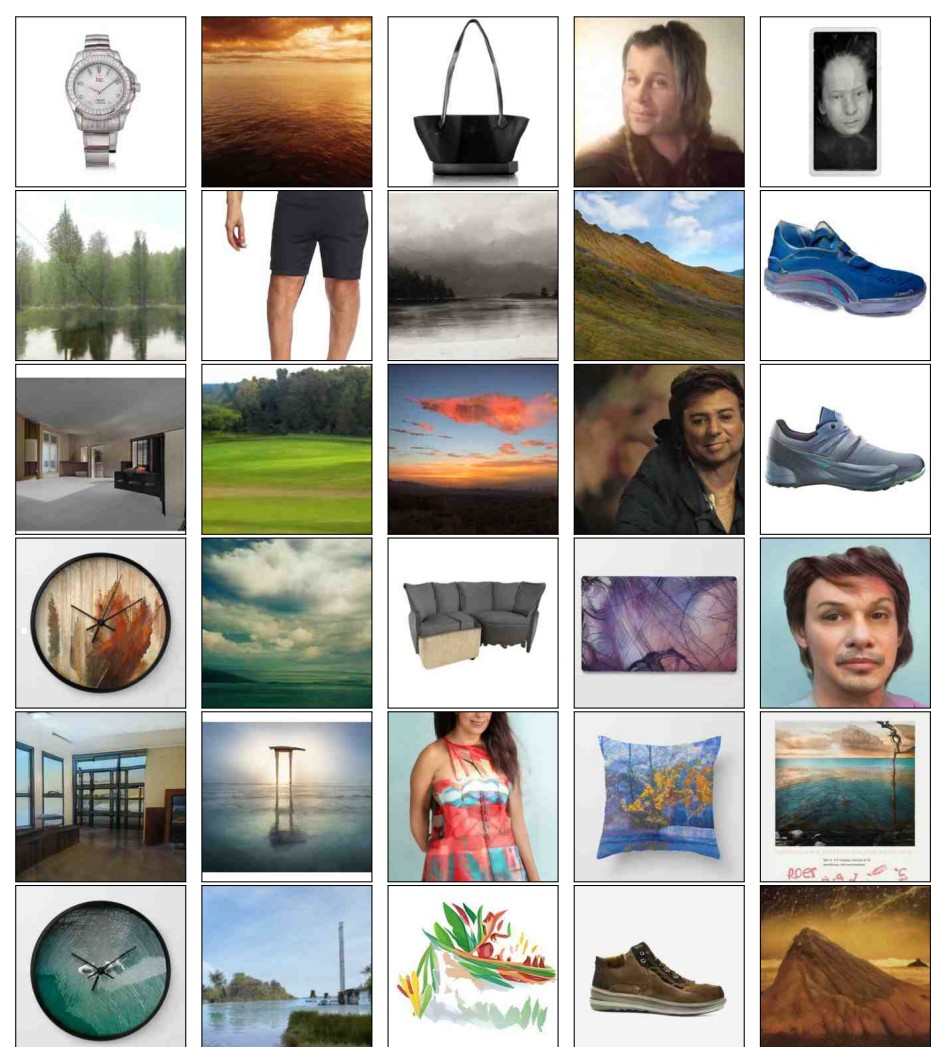

Figure 6: Unconditional generation by JPEG-LM.

## B   MORE QUALITATIVE EXAMPLES FROM JPEG-LM

In Figure 8, we show more JPEG-LM completions on partial images from FFHQ (zero-shot). Figure 6 and Figure 7 show further examples of JPEG-LM and VQ transformer on unconditional generation.

## C   MORE QUALITATIVE EXAMPLES FROM AVC-LM

More generations from AVC-LM can be found in Figure 9, Figure 10, Figure 11, Figure 12, and Figure 13. Similar to Figure 5, we observe realistic object movements (e.g., flag, clouds, clock, cars on the street, and camera movement towards a building).

## D   DETAILED CONFIGURATIONS FOR THE CANONICAL CODECS

Our JPEG encoding uses the `pillow` package. We specifically encode each image with: `image.save(format='JPEG', quality=25, subsampling="4:2:0", streamtype=2, restart_marker_blocks=1)`. More details about these arguments can be found at https://pillow.readthedocs.io/en/stable/handbook/image-file-formats.html#jpeg-saving. Our AVC/H.264 encoding uses the `ffmpeg` package. Specifically, the configurations/commands we

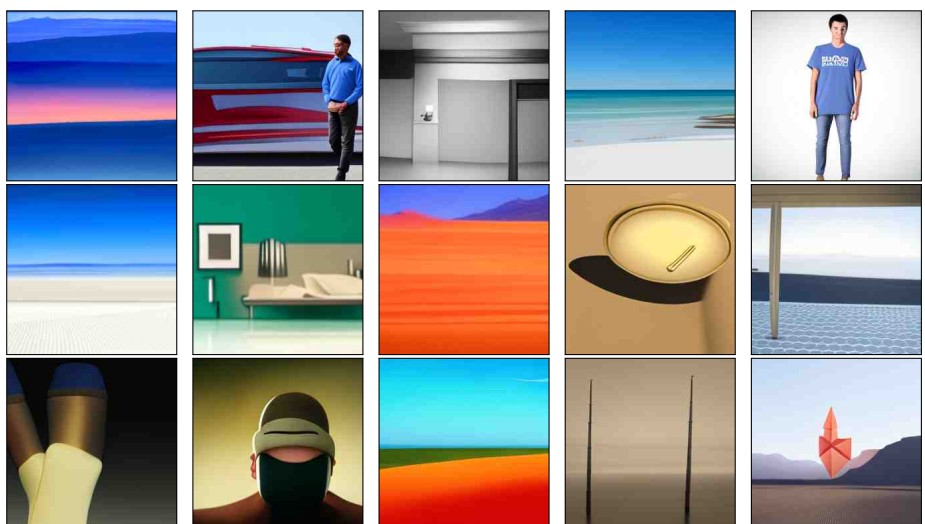

Figure 7: Unconditional generation by VQ transformer.

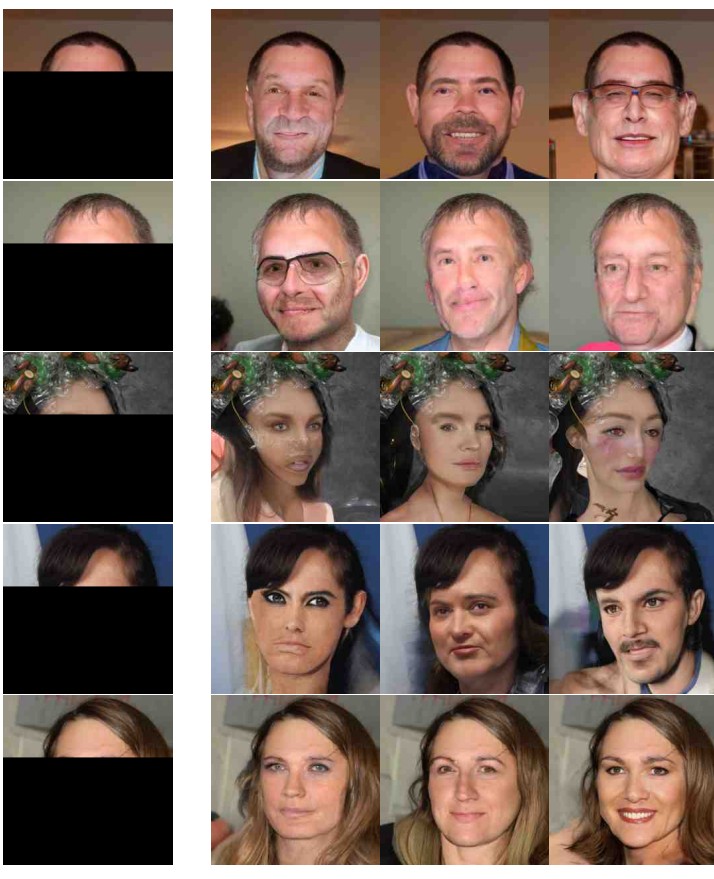

Figure 8: Generated images by JPEG-LM with partial FFHQ images as prompts (*zero-shot* generation). Similar to Figure 2, the generated facial expressions are modelled as JPEG bytes and mostly look realistic.

```
used are: ffmpeg -vf "fps=3,scale=256:144:force_original_aspect_ratio=decrease,
pad=256:144:(ow-iw)/2:(oh-ih)/2" -t 5 -c:v libx264 -pix_fmt yuv420p -profile:v
baseline -qp 37 -bf 0 -an -sn -x264opts "slice-max-mbs=1" -trellis 0 -me_method
```

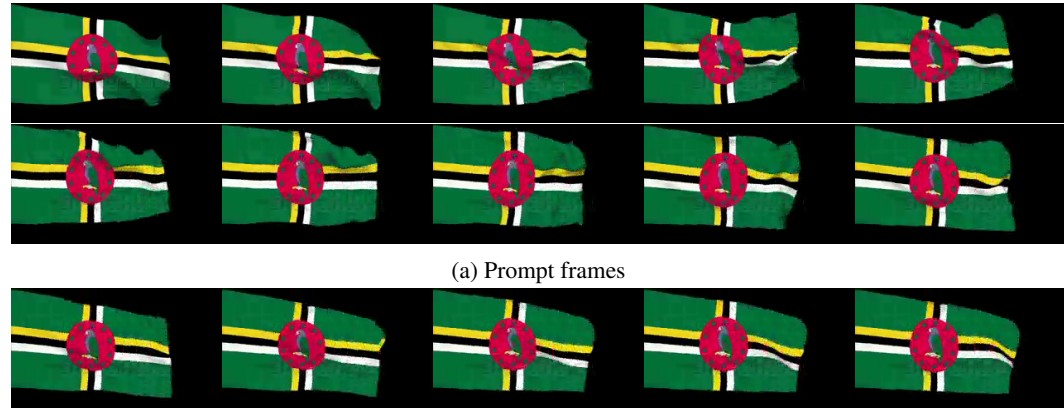

(a) Prompt frames

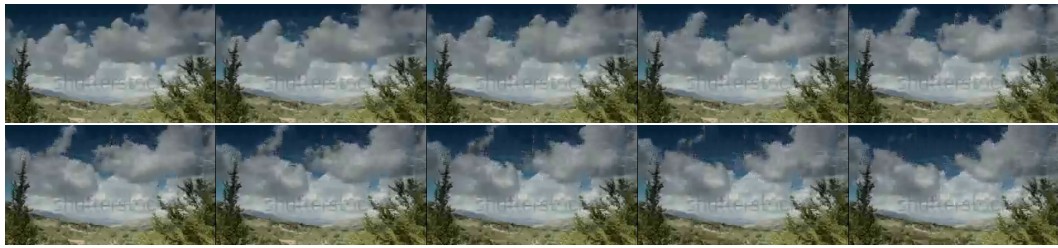

(b) Generated frames

Figure 9: Generated video frames by AVC-LM on held-out test data. The first 10 frames are given to the model as the prompt, and the last 5 frames are generated by the model.

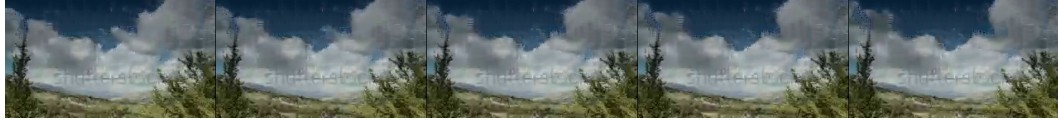

(a) Prompt frames

(b) Generated frames

Figure 10: Generated video frames by AVC-LM on held-out test data. The first 10 frames are given to the model as the prompt, and the last 5 frames are generated by the model.

```
dia -threads 1 -subq 0 -psy 0 -mixed-refs 0 -fast-pskip 0 -partitions none
-refs 3 -bsf:v h264_mp4toannexb. More details about these flags can be found at https:
//ffmpeg.org/ffmpeg.html.
```

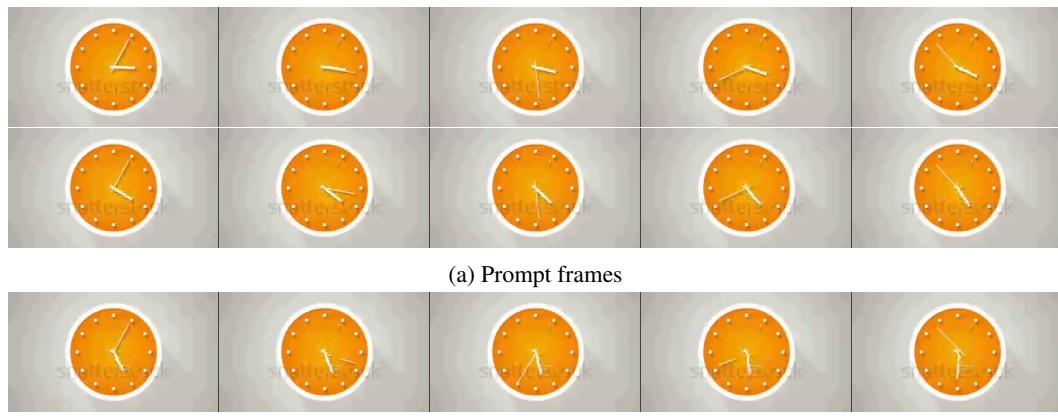

(a) Prompt frames

(b) Generated frames

Figure 11: Generated video frames by AVC-LM on held-out test data. The first 10 frames are given to the model as the prompt, and the last 5 frames are generated by the model.

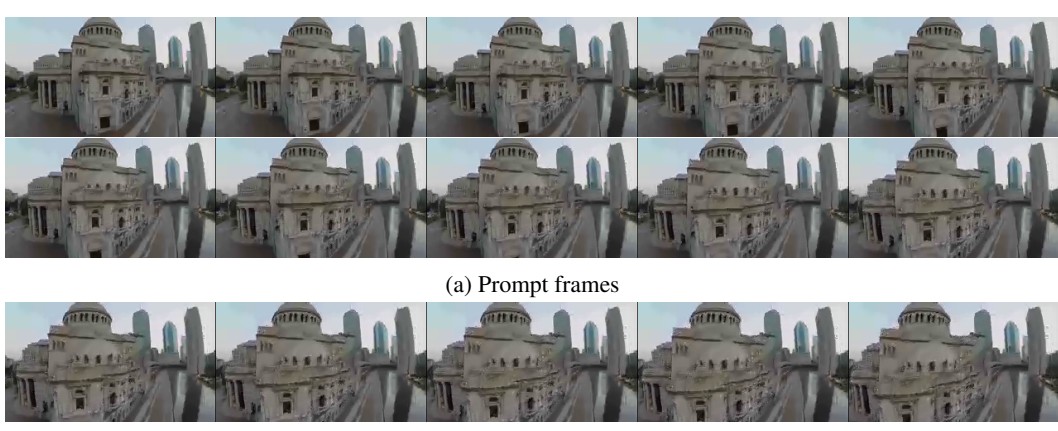

(a) Prompt frames

(b) Generated frames

Figure 12: Generated video frames by AVC-LM on held-out test data. The first 10 frames are given to the model as the prompt, and the last 5 frames are generated by the model.

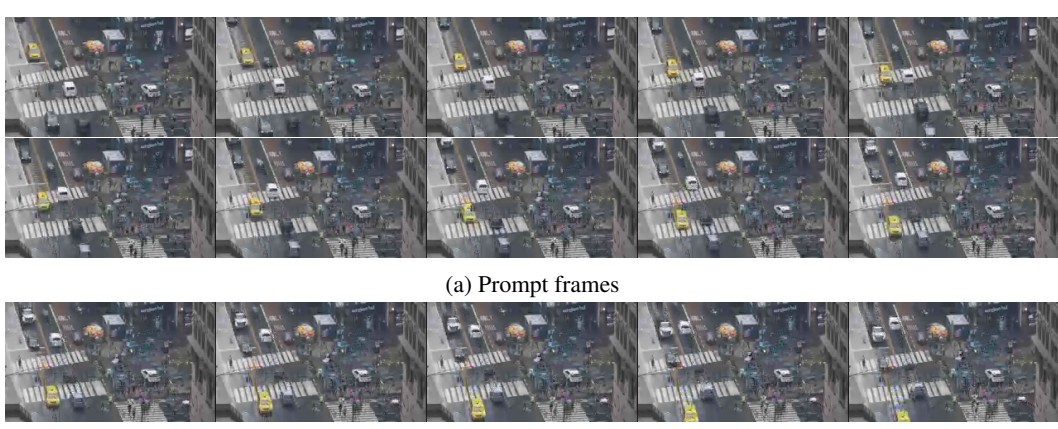

(a) Prompt frames

(b) Generated frames

Figure 13: Generated video frames by AVC-LM on held-out test data. The first 10 frames are given to the model as the prompt, and the last 5 frames are generated by the model.

