# OpenReview forum: "JPEG-LM: LLMs as Image Generators with Canonical Codec Representations"
_ICLR.cc/2025/Conference — Submitted to ICLR 2025_

### Official Review · Reviewer_VnZm · 2024-10-26

**Soundness:** 2
**Presentation:** 4
**Contribution:** 3
**Rating:** 5
**Confidence:** 5

**Summary:**

This paper introduces JPEG-LM, a novel autoregressive model for image and video generation using canonical codecs as representations. The authors aim to simplify the task of visual generation by eliminating the need for vision-specific modifications or complex discretization processes such as vector quantization (VQ). Their key contribution is the use of JPEG and AVC codecs to transform continuous visual data into discrete tokens, significantly reducing sequence length while maintaining generation quality. The models, based on a Llama-2 architecture, achieve state-of-the-art performance in generating realistic images and videos, especially outperforming VQ-based approaches in handling long-tail visual elements. Experiments demonstrate that JPEG-LM can generate high-quality visual content with fewer complications and show robustness in generating fine details like human faces or text.

**Strengths:**

Overall I find that the writing is clear, concise, and well-structured, making it easy for readers to follow the arguments and understand the key points. I like the novel idea of using canonical codecs to transform continuous images and videos into discrete tokens without the need for training VQ-VAEs. This opens up a promising direction of unified multimodal modeling using a joint transformer decoder.

**Weaknesses:**

* The method section of this paper lacks more technical details or background knowledge of the proposed method, especially for readers without much knowledge of canonical codecs. (Although I know the core idea of this paper is quite straightforward.)
* The biggest problem is that this paper only shows the inpainting results of image generation and preliminary results of video generation. It lacks standard results on benchmarks like ImageNet and COCO (Text-to-Image) to fully demonstrate the potential of the proposed framework. The only inpainting results are restricted to the "auto-regressive" manner, which deviates from conventional inpainting with arbitrary shapes and makes the comparison with diffusion models unfair. The paper claims to remove all visual-specific design to achieve unified framework with language models, but conducts no multimode experiments such as text-to-image generation, even though it uses data from LAION-5B.
* There is a similar work [1], which uses the same idea to model molecules.

[1] Flam-Shepherd, Daniel, and Alán Aspuru-Guzik. "Language models can generate molecules, materials, and protein binding sites directly in three dimensions as xyz, cif, and pdb files." arXiv preprint arXiv:2305.05708 (2023).

**Questions:**

* I really like the idea of this paper, but concerned about the experiments. If the above concerns can be addressed, I am willing to raise my score.
* I am not familiar with JEPG representation. Can it keep the spatial shape of an image? The paper claims not to use 2d RoPE, then how can it generate images/videos with different shapes or lengths?

---

> ### Author Response · Authors · 2024-11-27
>
> We appreciate Reviewer VnZm for their insightful and helpful comments!
>
> ====
>
> ## Re: Background introduction to canonical codecs.
>
> We thank the reviewer for this feedback! We will include an additional section or paragraph discussing the background of canonical codecs, especially on the standard JPEG procedures of discrete cosine transform, quantization, run-length encoding and entropy coding.
>
> ====
>
> ## Re: Inpainting image generation.
>
> In addition to the inpainting/conditional image generation setup, we also test on unconditional image generation (e.g., with experiments in Table 3). As for not training a class or text conditioned image generation model, it is a deliberate choice at the beginning of our project. We want to build a pure model on JPEG bytes only, (1) to highlight the ablated effect of using different data representations for pure visual modeling, and (2) with a future vision that a pure JPEG model can be tuned together with pure text/action/other-modality models in a next stage (since they would usually share the same LLM architecture) --- if we pre-model image classes in the model, it may hinder future adaptation where class-conditioning is not desired. In short, we want to model the more general and flexible p(x) instead of p(x|y) in this work. We will add a discussion on our design choice here and on potential future work that can straightforwardly tune pure text and pure image model together for cross-modality conditioned generation (e.g., Liang et al., 2024 -- https://arxiv.org/abs/2411.04996).
>
> ====
>
> ## Re: Similar work on modeling molecules.
>
> We thank the reviewer for mentioning this work on modeling molecules/materials/protein. We will add it to our related work section and discuss the application of such relevant methods in fields outside vision and language.
>
> ====
>
> ## Re: Spatial shape of JPEG images.
>
> The spatial shape of the JPEG images can be specified in the header of the file. Though in the main experiments of the work, we generate with a fixed resolution of 256x256, in one of our pilot studies, we tried with multiple-resolution images, and the JPEG language model is capable of generating different resolution images. This is achievable because (1) the generation can be conditioned on the JPEG header describing the target resolution, and (2) JPEG can use special separator tokens between each 16x16 patches in our data (a standard to prevent errors and byte corruptions from flowing to other patches) and subsequently count them to keep track of the current progress in generation. Also, in JPEG's grammar, byte patterns like "FFDA" and "FFD9" are essentially BOS and EOS tokens in LLMs, so the model can learn to end a sequence early or late by the timing of outputting the "FFD9"/EOS token.

---

> > ### Comment · Reviewer_VnZm · 2024-11-27
> >
> > Thanks for your response. Without any experiments or further revisions, this rebuttal does not address my major concerns. So I will keep my score.

---

### Official Review · Reviewer_hMNX · 2024-10-29

**Soundness:** 3
**Presentation:** 3
**Contribution:** 3
**Rating:** 6
**Confidence:** 4

**Summary:**

In this paper, the authors propose a new auto-regressive model that directly generates compressed binary data in JPEG or AVC formats. As such canonical codec provides compressed discrete representation of data, it allows the model to simply follow the standard architecture of language models without any vision-specific modification. The experimental results show that the proposed model outperforms the baseline method that adopts VQ-VAE for tokenization, especially when less common contents appear in the image.

**Strengths:**

- The main research question is quite simple but really interesting: can LLM-like models directly generate compressed binary data in an efficient and effective manner? The experimental results shown in this paper provide a positive answer to this question, which would be worth reporting in the community.

- The design of the proposed model is also simple. It follows LLaMA-2 model architecture, and its vocabulary is basically a simple byte one slightly extended by byte-pair encoding to handle a few frequently appearing sequences in metadata and headers. This simplicity enhances reproducibility of the method and is suitable as a baseline for future studies.

- The experimental results show that the proposed model outperforms VQ Transformer using learning-based discrete representations obtained by VQ-VAE, which follows a popular design in recent studies. They would let us rethink how to effectively compress data for generative models.

**Weaknesses:**

- The JPEG images for training were compressed with a very low quality factor of 25 (compared with 70-90 in standard usage), which causes many artifacts, such as ringing and block noise, in the generated images as shown in Section 5 and appendix. I conjecture that this choice aims to substantially reduce the number of tokens processed by the proposed method. It would be beneficial if the authors could provide some justification on this setting.

**Questions:**

- Model architecture
  - Does "no 2D positional embedding" indicate that the proposed model simply adopts the standard rotary positional embedding from the default setting of LLaMA2?

- Experiments
  - Is there any possibility that the proposed model generate collapsed binary data (which is impossible to decode)? If yes, how did the author treat such data in the experiments?
  - Were the test images used for prompts and FID computation also encoded by JPEG with quality=25? The results of VQ and ImageGPT shown in Fig. 2 seem to contain some ringing artifacts and block noise, which should not occur without JPEG encoding. Could you clarify the reason for these artifacts?

---

> ### Author Response · Authors · 2024-11-27
>
> We appreciate Reviewer hMNX for their positive review and constructive feedback!
>
> ====
>
> ## Re: Reason for using low quality factor in JPEG.
>
> Yes, the main reason for using a low quality factor of 25 in JPEG is to reduce the overall sequence length (in contrast to the ImageGPT case that uses raw pixel information without much compression). The sequence length manageable by standard transformers indeed creates a limiting tradeoff in our case (more manageable sequence length would lead to more visual artifacts), but we hope the issue can be mitigated by newer designs and architectures in language modeling transformers that enable longer contexts in the future. The purpose of this work is to show the simplicity and feasibility of modeling canonical compression codecs in a generative setup with language modeling architectures. Future work can explore more advanced and powerful codecs (e.g., neural image/video codecs that also perform variable-length compression for data transmission) than the very old JPEG algorithm, following the positive results shown in our work.
>
> ====
>
> ## Re: Llama-2 positional embedding.
>
> Yes, we use the standard RoPE positional embedding from the default Llama-2 setup (no structural modifications are made to the Llama-2 architecture at all). This simple and straightforward design choice works well and can be generalizable to other modalities in the future.
>
> ====
>
> ## Re: Possibility of generating invalid binary data.
>
> Theoretically, it is totally possible to generate corrupted binary JPEG data (like in NLP, generating grammatically incorrect language). The JPEG decoding algorithm would still be able to decode corrupted data, where the patch containing invalid data is shown with obviously broken color or a fully gray/chessboard-like pattern. We almost never observe such corruption in the generations of our trained JPEG-LM (in fact, at the very early stage of our model training, we already stopped seeing corrupted generation, indicating the model learns valid JPEG grammar first and then the underlying semantics). In the rare cases where corrupted binary data is generated, such corrupted data is included and reflected in our overall evaluation results as well.
>
> ====
>
> ## Re: Details about prompts, FID computation, and qualitative examples.
>
> Only the prompts to the JPEG-LM model will be encoded by JPEG with a quality factor of 25. Prompts to all other models will be in the original format (e.g., would be unfair to give VQ transformers a prompt with the JPEG low quality factor). Interestingly, the original format itself can sometimes be .jpg as well (since it's a universal codec/file format) but definitely with much higher quality (usually in the 70-90 range). Also, our FID computation involves gold images and model generations in original quality for a fair comparison. We see artifacts in the qualitative examples of non-JPEG models (Figure 2) only because when we export the images for the qualitative show, we have a legacy setting with and without JPEG low quality save for our own inspection. The shown qualitative examples in Figure 2 are with the low quality export, but we will replace them with high quality export in the next revision. We thank the reviewer for their attention to details.

---

> > ### Comment · Reviewer_hMNX · 2024-11-27
> > **Thanks for the response.**
> >
> > Thanks for your response.
> >
> > > Re: Reason for using low quality factor in JPEG.
> >
> > I somewhat agree with the authors' argument but it would be beneficial to discuss how challenging it is to train the proposed model with a standard quality factor in JPEG, as one of limitations. Specifically, if we set the quality factor to be around 70-90, how long would the sequence be? (and how longer is it compared with that in a standard-size LLM?)
> >
> > > Re: Llama-2 positional embedding.
> >
> > Thanks, I got it.
> >
> > > Re: Possibility of generating invalid binary data.
> >
> > Differently from the case of standard text generation, how to handle corrupted data in the evaluation would be crucial, because the corrupted data cannot be decoded into images, thus cannot be directly used to compute any metrics for image-quality evaluation (e.g., FID). As this issue particularly appears in the proposed method, I encourage the authors to clearly describe the details on how such data is processed in the evaluation.
> >
> > > Re: Details about prompts, FID computation, and qualitative examples.
> >
> > Thanks for the clarification. As the current figure does not provide accurate qualitative comparison, I would like to see the revised manuscript.

---

> > > ### Author Response · Authors · 2024-12-03
> > > **Thanks for the follow-up questions.**
> > >
> > > We appreciate Reviewer hMNX for their follow-up questions!
> > >
> > > ### Re: Re: Reason for using low quality factor in JPEG.
> > >
> > > Our average sequence length would be 8.7K when quality factor=70, and the average sequence length is 16K when quality factor=90 (in comparison, the average sequence length is 4.3K when quality factor=25). Our work uses the Llama-2 architecture (extended the max sequence length to 12K to prevent fragmentation during data packing). As a reference, the pretrained Llama-2 models for texts have a sequence length of 4K (for the more recent Llama-3 models, the pretraining sequence length is 8K). We hope these provide more contexts for our design choice, and we will include these as a part of our discussion section.
> > >
> > > ### Re: Re: Possibility of generating invalid binary data.
> > >
> > > We would like to clarify that there are two types of corruption:
> > >
> > > (1) Viewable corruption. As mentioned in our previous response, we may see (with very low chance) corrupted patches with obviously broken color or a fully gray/chessboard-like pattern. Such corruption still leads to viewable images. As a dummy example, in our Figure 2 Column (a), these prompt images can be considered as "corrupted", since the bytes corresponding to the lower half of the images are directly removed. When exporting the image, the JPEG decoder fills in the image with gray patches (when the data is not completely removed but flipped a bit or so, the patches might show broken color or chessboard-like patterns). The image is still viewable in the case and has the right size since the headers at the beginning of the sequence specify it (and in cases the sequence does not end itself, we append two bytes FFD9 to the end since according to the standard JPEG grammar). Images with such (viewable) corruption are compatible with the FID evaluations.
> > >
> > > (2) Unviewable corruption. The corrupted data cannot even be viewed/opened by a JPEG decoder/file reader. An imagined case of such kind is throwing a random sequence of bytes to the JPEG file reader. However, we never encountered such unviewable corruption. The top-level JPEG grammar is [FFD8, headers, FFDA, compressed data (optionally with FFD0-FFD7 separators to limit error flowing between patches), FFD9]. Our JPEG headers are so easy to learn (they are almost identical even for images with different resolution). The quantization tables and Huffman tables are fixed and injected as default. The corruption that happens inside the "compressed data" part is viewable. And we never have the unviewable corruption in our evaluation.
> > >
> > > We think the reviewer might be referring to the latter type, unviewable corruption, so we just want to make the clarification that it never happened and we were referring to the former type, viewable corruption. We will include this discussion in our next version of the paper.

---

### Official Review · Reviewer_U4DX · 2024-10-31

**Soundness:** 4
**Presentation:** 4
**Contribution:** 4
**Rating:** 6
**Confidence:** 5

**Summary:**

The paper introduces JPEG-LM and AVC-LM, two autoregressive language models that generate images and videos using standard codecs (JPEG for images and AVC/H.264 for videos) instead of specialized, vision-specific tokenization or pixel-based sequences. By directly modeling compressed byte sequences, these models avoid complex pre-processing or training processes typical of vector quantization (VQ) models. JPEG-LM, trained on JPEG image bytes, consistently outperforms VQ-based models in generating realistic visuals and demonstrates particular robustness in long-tail visual elements, such as small facial features and text characters.

**Strengths:**

1. Efficiency and Simplicity: JPEG-LM and AVC-LM bypass the need for complex tokenization and preprocessing by using JPEG and AVC codecs, which simplifies training compared to VQ-based methods.

2. Performance on Long-Tail Elements: The approach excels in generating visually complex, less frequent elements (like small text or detailed faces) where VQ models often degrade in quality.

3. Versatility in Modeling: Using a standard LLM architecture without vision-specific adjustments, JPEG-LM extends the utility of language models to image and video generation effectively, aligning with LLMs for text.

4. Robustness in Out-of-Distribution Scenarios: The codec-based approach is less sensitive to distributional shifts, potentially due to its reliance on stable, non-neural compression methods.

**Weaknesses:**

1. Limited Improvements Over Diffusion Models: While JPEG-LM shows promise, its performance does not match the leading results from diffusion-based models, especially in complex or higher-resolution tasks.

2. Dependency on JPEG Compression Constraints: Although effective, relying on JPEG and AVC formats may limit the granularity of generated details, especially in nuanced textures or high-frequency patterns.

3. Scalability for Video Generation: While AVC-LM is a compelling proof of concept, its application is limited to low frame rates and resolutions, indicating potential constraints in real-world, high-resolution video synthesis.

**Questions:**

1. How does JPEG-LM handle artifacts introduced by JPEG compression, and would incorporating a post-processing step improve image quality?

2. Could this codec-based approach extend effectively to other file formats, such as PNG for lossless compression or HEVC for higher-definition video?

3. Have you considered integrating JPEG-LM with diffusion techniques to leverage the best of both methods for higher fidelity in complex textures?

---

> ### Author Response · Authors · 2024-11-27
>
> We thank Reviewer U4DX for their positive feedback!
>
> ====
>
> ## Re: Limited performance improvements compared to diffusion-based models.
>
> We showed a performance comparison with diffusion-based models as secondary evaluations in Table 1 and 2 and showed matching or better performance of JPEG-LM (e.g., on zero-shot ImageNet when r_prompt=0.75, JPEG has a FID of 34, much lower/better than SD/VQ Diffusion's 58/57; on zero-shot FFHQ when r_prompt=0.5, JPEG has a FID of 27, much lower/better than SD/VQ Diffusion's 89/40). In addition, our focus and main experiment in this work is actually a controlled comparison with the autoregressive VQ transformer, since we are interested in whether regular LLMs can effectively learn visual generation simply by modeling JPEG file bytes.
>
> ====
>
> ## Re: Limitation in nuanced textures and high-frequency patterns.
>
> At a high level, JPEG compresses the nuanced textures and high-frequency patterns more since human eyes are less sensitive to them. All compression would involve losses of information, and such canonical compression is both interpretable and robust to out-of-distribution data (e.g., as shown in our Figure 4).
>
> ====
>
> ## Re: Post-processing of JPEG generation, other file formats, and incorporation of diffusion techniques.
>
> These would be great future work directions! For example, exploring the correlation between different codec complexity and the effectiveness of generative modeling, post-processing JPEG generation with upsampling diffusion, etc. We believe our work is an interesting and important first step to these promising research questions.
>
> ====
>
> We'll include additional discussion for all above, and we thank the reviewer for the positive comments!

---

### Official Review · Reviewer_YdAt · 2024-11-02

**Soundness:** 3
**Presentation:** 3
**Contribution:** 2
**Rating:** 8
**Confidence:** 3

**Summary:**

The article uses compression algorithms like JPEG to encode image data into discrete tokens, then leverages current large language models to learn from these encoded tokens. The advantage of this approach is that it avoids artifacts and distortions often introduced by some existing image tokenizers during reconstruction. I find this innovation compelling, yet I have significant concerns that make me cautious about these issues.

Firstly, JPEG has a limited compression ratio under visually lossless conditions. Meanwhile, DNN-based tokenizers are continuously improving, with better reconstruction quality and increasing compression ratios. Additionally, I didn’t find any convincing comparative experiments in the article, such as comparisons with works like Llamagen or VAR. I’ve given it a score of 6, but if you can address my concerns, I will revise my score.

**Strengths:**

1. This article is well-organized and easy to understand.

2. I trust the authors' experiments; they look very interesting and demonstrate that JPEG compression, along with other compression algorithms, can be applied to image compression, decompression, and image/video generation.

3. The article shows substantial innovation. I believe it’s a good paper.

**Weaknesses:**

1. This article lacks many necessary experiments, such as comparisons with VAR and Llamagen.

2. I have doubts about the scalability of this approach, as JPEG has a limited compression ratio. Even with a 4-10x compression ratio (where higher compression ratios often lead to low-quality generated content), it may still be insufficient to support long-duration video generation.

3. Generating high-quality images may require more time.

**Questions:**

1. Why wasn’t a comparison made with Llamagen and VAR?

2. Why wasn’t Llama 3 used?

3. The metrics seem unreasonable—why wasn’t there conditional image generation included?

---

> ### Author Response · Authors · 2024-11-27
>
> We appreciate Reviewer YdAt for their insightful feedback and positive comments!
>
> ====
>
> ## Re: Comparison with VAR and LlamaGen.
>
> We thank the reviewer for bringing up these two related works.
>
>
> For VAR, the underlying idea is related, but VAR’s contribution is orthogonal to our work’s contribution. VAR aims to improve the LLM aspect of autoregressive next-token prediction with partially autoregressive token prediction across multiple data scales/resolution (still consists of VQ tokens), whereas JPEG-LM aims to improve the core representation aspect of VQ with JPEG compression. I.e., VAR cannot be used as baseline to our JPEG-LM, but in the future, our autoregressive JPEG-LM may benefit from the next-scale/next-resolution prediction in VAR, and can interestingly be extended to a next-compression prediction setup, where the mostly compressed image is generated first, followed by the generation of less compressed images. We will include this discussion in our revision.
>
>
> For LlamaGen, the underlying idea closely follows previous work on VQ tokenization and autoregressive modeling with transformers and investigates the scaling and data quality aspects (e.g., a use of proprietary high-quality data). From a modeling perspective, our VQ transformer baseline already shares a similar setup but with a special focus on controlled experiments with JPEG-LM on model training and data configuration. In addition, all of our model training and paper are concurrent to the LlamaGen work. Future explorations may investigate a controlled comparison of Transformers with JPEG encoding and with LlamaGen's VQ image tokenizer (a new model training process of which we lack time and resources during the rebuttal period unfortunately), in addition to our existing controlled comparison of Transformers with JPEG and a strong open-source VQ tokenizer from another prior work (Tang et al). We will discuss all of the above in our related work section.
>
> ====
>
> ## Re: JPEG compression ratio is limited and may be insufficient for future work like video generation, whereas DNN-based tokenizers are improving in the long term.
>
> We would like to highlight that while the DNN-based neural tokenizers may be improving continuously in the future, neural tokenizers will continue to be trained to fit certain predetermined data distributions. If people care about the long-tail and out-of-distribution use-cases, canonical codecs like JPEG can still have advantages due to its interpretable and robust compression (as shown in our Figure 4). Another important purpose of this work is to show that compressive codecs for variable-length data transmission are indeed feasible to be used directly in visual generation. Following the intuition of this work, future work in video generation may explore existing neural video codecs as a promising alternative to the mainstream VQ video tokenizers.
>
> ====
>
> ## Re: Potential decoding time for high-quality images.
>
> For very high resolution images, we believe research in upsampling would be useful in addition to our work. For example, using diffusion to upsample our moderate-resolution generation. The VAR work mentioned by the reviewer can also be a good approach to combine. We will include this relevant discussion in the next draft.
>
> ====
>
> ## Re: Llama-3 architecture.
>
> Our main training and experiments were done before Llama-3's release. However, since we trained JPEG-LMs from scratch (only used Llama-2's architecture but not weights) and Llama-2 and 3 share a very similar architecture, we believe using Llama-3 would lead to a similar result. The selection of the LLM architecture is a rather general choice. We also have pilot studies (though out of the scope of this paper) that use the Mamba architecture for JPEG-LM and are showing initial successes.
>
> ====
>
> ## Re: Class or text conditioned generation.
>
> This is a deliberate choice at the beginning of our project. We want to build a pure model on JPEG bytes only, (1) to highlight the ablated effect of using different data representations for pure visual modeling, and (2) with a future vision that a pure JPEG model can be tuned together with pure text/action/other-modality models in a next stage (since they would usually share the same LLM architecture) --- if we pre-model image classes in the model, it may hinder future adaptation where class-conditioning is not desired. In short, we want to model the more general and flexible p(x) instead of p(x|y) in this work. We will add a discussion on our design choice here and on potential future work that can straightforwardly tune pure text and pure image model together for cross-modality conditioned generation (e.g., Liang et al., 2024 -- https://arxiv.org/abs/2411.04996).

---

### Meta-Review · Area_Chair_fub8 · 2024-12-21

**Metareview:**

This paper proposes to apply an autoregressive Transformer to directly generate the bytes in the JPEG representation of images and videos. The main motivation is to replace the role of trained VQVAEs which simplifies the modeling pipeline. Reviewers find the idea interesting, and most ratings are positive except for reviewer VnZm. After reviewing the paper and discussions, the AC believe's that the concerns raised by VnZm is not addressed. The fact that the paper mainly relies on image completion tasks as an evaluation makes it difficult to support the claim that the JPEG codes are indeed superior than VQVAEs. Although Table 3 includes uncondtional sampling evaluations, their absolute numbers seem very high for both VQVAE baseline and the proposed method. I would strongly suggest the authors to consider standard image generation benchmarks and evaluation protocols, such as class conditional/unconditional sampling with standard FID evaluation protocols. The other issue is that I find the use of 'LLM' misleading -- one would have assumed that it utilizes a pretrained language model, whereas in reality it's just an autoregressive Transformer trained from scratch. I would suggest clarifying this by either changing the title or explaining this explicitly early on. Based on these considerations, I don't feel like this paper is ready to be accepted, but I hope the authors can keep improving it.

**Additional Comments On Reviewer Discussion:**

Reviewers find the idea interesting, and most ratings are positive except for reviewer VnZm. After reviewing the paper and discussions, the AC believe's that the concerns raised by VnZm is not addressed.

---

### Decision · Program_Chairs · 2025-01-22

Reject